# The Role of Platelet-Rich Plasma on the Chondrogenic and Osteogenic Differentiation of Human Amniotic-Fluid-Derived Stem Cells

**DOI:** 10.3390/ijerph192315786

**Published:** 2022-11-27

**Authors:** Alessio Giannetti, Andrea Pantalone, Ivana Antonucci, Sandra Verna, Patrizia Di Gregorio, Liborio Stuppia, Vittorio Calvisi, Roberto Buda, Vincenzo Salini

**Affiliations:** 1Department of Life, Health and Environmental Sciences, University of L’Aquila, 67100 L’Aquila, Italy; 2Clinic of Orthopaedics and Traumatology, “SS. Annunziata” Hospital, 66100 Chieti, Italy; 3Center of Advanced Studies and Technology (CAST), “G. D’Annunzio” University of Chieti-Pescara, 66100 Chieti, Italy; 4Immunohematology and Transfusional Medicine Service, “SS. Annunziata” Hospital, 66100 Chieti, Italy; 5Department of Medicine and Aging Sciences, “G. D’Annunzio” University of Chieti-Pescara, 66100 Chieti, Italy; 6Department of Orthopaedics and Traumatology, San Raffaele Hospital, 20132 Milan, Italy

**Keywords:** amniotic fluid, stem cells, PRP, orthobiology, cartilage, regenerative medicine

## Abstract

Amniotic fluid represents a new and promising source of engraftable stem cells. The purpose of this study was to investigate the in vitro effects of platelet-rich plasma (PRP) on amniotic-fluid-derived stem cells (AFSCs) on chondrogenic or osteogenic differentiation potential. Amniotic fluid samples were obtained from women undergoing amniocentesis for prenatal diagnosis at 16–18 weeks of pregnancy. Undifferentiated human AFSCs were cocultured with PRP for 14 days. The study includes two protocols investigating the effects of activated PRP using two different methods: via freeze–thaw cycles and via the addition of calcium gluconate. On the 14th day of culturing, the differentiation potential of the cocultured AFSCs was then compared with undifferentiated AFSCs. Staining with alcian blue solution (ABS) and alizarine red solution (ARS) was performed, and chondrogenic- and osteogenic-associated genes markers were investigated. ABS demonstrated enhanced glycosaminoglycan expression. Cocultured cells expressed chondrocyte-associated genes, determined by real-time polymerase chain reaction (RT-PCR), including type I collagen, type II collagen, COMP, and aggrecan. In regard to the osteogenic markers, osteopontin and bone sialoprotein, there were no changes. In particular, the activation of PRP using the freeze–thaw cycle protocol showed a higher expression of the chondrogenic markers. Our preliminary in vitro results showed that PRP has good potential in the chondrogenic differentiation of AFSCs.

## 1. Introduction

Osteoarthritis (AO) is currently one of the most frequently diagnosed chronic diseases, affecting an estimated 10% of men and 18% of women over 60 years of age, and with the increase in life expectancy, both its prevalence and incidence are expected to rise. This condition is degenerative until the loss of function and disability leads to important healthcare and social costs [1,2]. In addition to chondral pathologies, skeletal diseases, such as nonunion and bone defects due to trauma, infections, or tumors, represent a great challenge for orthopedic surgeons. 

Bone, in fact, is the second-most transplanted tissue worldwide after blood, with over two million bone grafting surgeries being conducted per year [3].

Therefore, the treatment of traumatic or degenerative osteochondral defects represents one of the main targets of regenerative medicine in order to replace damaged tissues [4].

Several therapeutic strategies are currently available, but the combined use of platelet-rich plasma (PRP) and mesenchymal stem cells (MSC) is an alternative that is increasingly being adopted by physicians, providing encouraging results for orthobiology.

### 1.1. Platelet-Rich Plasma

PRP is a blood-derived concentrate with a platelet concentration several times above the baseline. PRP was discovered in 1914 when it was prepared for intravenous transfusion, and it was used only for the hemostatic properties of the platelets until 1974, when Ross et al. [5] observed that activated platelets promoted proliferation in monkey arterial smooth muscle cells in vitro.

Platelets, when activated, release 95% of the chemokines, cytokines, and growth factors (GFs) contained in the α-granules within 1 hour; then, the release of the remaining content reaches a plateau, and a slow secretion continues for up to 7–8 days.

The most important GFs are platelet-derived growth factor (PDGF), transforming growth factor-beta (TGF-beta), fibroblast growth factor (FGF), insulin-like growth factor-1 (IGF-1), connective tissue growth factor (CTGF), epidermal growth factor (EGF), and hepatocyte growth factor (HFG), providing the PRP with chemoattractive, angiogenic, proliferative, and proregenerative properties. In addition, acting as a hydrogel, activated PRP could be suitable as a cell-delivery vehicle in the context of tissue engineering [6].

Because of all these characteristics, PRP has found wide application in clinical practice, useable as local infiltration in many orthopedic diseases, such as as mild osteoarthritis, acute and chronic tendinitis, and plantar fasciitis, as well as in pathologies that are not strictly orthopedic, such as the treatment of chronic wounds and diabetic ulcers, and it has even found application in other fields of medicine, such as plastic surgery and medical esthetics [7,8,9,10,11,12].

Furthermore, different microRNAs involved in mesenchymal tissue regeneration are also present in platelets’ microvesicles, and some of them, such as microRNA-23b, have been hypothesized as being strictly implicated in the differentiation of MSC into chondrocytes [13].

The practice of combining MSC and PRP in regenerative medicine and, of course, in orthobiology in the treatment of osteochondral defects continues to be an area of interest for many investigators. 

### 1.2. Mesenchymal Stem Cells + Platelet-Rich Plasma

The abilities of MSCs, including self-renewal and multilineage differentiation into other types of cells, in addition to their analgesic, immunomodulatory, and anti-inflammatory properties, offer encouraging strategies for replacing or regenerating damaged tissues [14]. Among the MSCs, bone marrow stem cells (BMSCs) have certainly been the most used and the most commonly investigated cells in regenerative medicine [15,16]. On the other hand, adipose-derived stem cells (ADSCs) have the advantage of being more numerous and easier to harvest [17]. Nevertheless, MSCs can be harvested from many other sources, such as peripheral blood [18], the lungs [19], the synovial membrane [20], dental pulp [21], satellite muscle cells [22], the placenta [23], and the umbilical cord [24].

In a review of the international literature, it emerges that promising results have been obtained by combining MSC and PRP, both in vitro [22,25,26] and in vivo [18,21,22,24], and more and more clinical trials are reporting encouraging results in the most common orthopedic pathologies, such as mild arthritis, tendinitis, and muscle lesions, showing improvements in patient-reported outcome measures (PROMs) [7,27].

### 1.3. Amniotic-Fluid-Derived Stem Cells

The amniotic membrane has been known for its clinical use and has already been investigated in different fields for applications [28], while less is known about amniotic fluid. Recently, it was found that amniotic fluid could represent a promising new source for harvesting stems cells with therapeutic applications [4,29,30]. Amniotic-fluid-derived stem cells (AFSCs) are collected via amniocentesis in women undergoing prenatal diagnosis (16th–18th week of pregnancy). These cells possess interesting characteristics; in fact, AFSCs present intermediate properties, placing them between embryonic stem cells (ESCs) and adult stem cells (ASCs) in terms of their capabilities. Indeed, AFSCs present a greater potency than ASCs. In addition, in a comparison between ESCs and AFSCs, the latter are easier to collect and more ethically acceptable because no embryo needs to be suppressed. In addition, they are more genetically stable, and therefore, they do not induce teratomas after in vivo transplantation [4,29,30].

Despite the important role of PRP in orthobiology, the literature currently lacks studies on the potential of PRP in the chondrogenic and osteogenic differentiation of AFSCs. The purpose of the study was to investigate the in vitro effects of PRP on the potential of AFSCs to differentiate.

## 2. Materials and Methods

### 2.1. Preparation of Platelet-Rich Plasma

The PRP was obtained from a pint of whole blood, taken from a volunteer periodic blood donor and considered suitable for blood donation. The platelet count from the complete blood count (CBC) was 325 × 10^3^ platelets/µL.

Then, 450 cc of the whole blood was collected in a quadruple bag and centrifugated (Jouan KR4i centrifuge). The four bags are connected. Welding was performed to separate them so as to maintain the sterility of the products.

A first centrifugation (1800 rpm for 10 min at room temperature) was performed to separate the red cell concentrate from the PRP. The PRP was centrifugated in the same centrifuge at 3500 rpm for 15 min. At this stage, two bags had been obtained, one containing the PRP, and the other containing the platelet-poor plasma.

The CBC for the PRP was 1.2 × 10^6^ platelets/µL. The PRP was 10% of the volume of the original whole blood. In the end, the PRP was aliquoted in vials under a sterile hood and stored at −40 °C for a maximum period of 3 months. Sterility was checked on two random aliquots, with negative results. The PRP aliquots were thawed at 37 °C at the time of use.

### 2.2. Amniotic-Fluid-Derived Stem Cells

#### 2.2.1. Isolation and Culture

Amniotic fluid samples were obtained from women undergoing amniocentesis for prenatal diagnosis at 16–18 weeks of pregnancy after written informed consent had been obtained. For each sample, 2–3 mL of amniotic fluid, corresponding to a cell number ranging from 2 × 10^3^ to 2 × 10^6^ was centrifuged for 10 min at 1800 rpm. Pellets were resuspended in Iscove’s Modified Dulbecco’s Medium (IMDM), supplemented with 20% fetal bovine serum (FBS), 100 U/mL penicillin, 100 μg/mL streptomycin (Sigma), 2 mM L-glutamine, and 5 ng/mL basic fibroblast growth factor (FGF2) and incubated at 37 °C with 5% humidified CO_2_. 

After 7 days, nonadherent cells were removed, and the adherent cells were allowed to grow in the same medium, which was changed every 4 days. In this way, the AFSCs were isolated from the original cell population and expanded in a culture until the third passage (Figure 1).

#### 2.2.2. Study Protocols

The study included two protocols to investigate the effects of PRP activation via different methods. The medium of Protocol 1 was: IMDM (90%), PRP (10%), L-glutamine (200 mM), and penicillin/streptomycin (100×). In this protocol, the PRP was activated using freeze–thaw cycles.

On the other hand, the medium of Protocol 2 was: IMDM (87%), PRP (10%), calcium gluconate (3%), L-glutamine (200 mM), and penicillin/streptomycin (100×). In this case, the PRP was activated via the addition of calcium gluconate.

Undifferentiated human AFSCs were cocultured, with the medium studied for 14 days. The medium was changed every 2 days.

The two protocols were compared, with a cell lineage of undifferentiated AFSCs used as the control group (control medium: IMDM (90%), FBS (10%), L-glutamine (200 Mm), and penicillin/streptomycin (100×)).

#### 2.2.3. Staining

On the 14th day of culturing, two different stains were investigated: one using alcian blue solution (ABS), which binds sulphurated acid mucins and glycosaminoglycans, and one using alizarine red solution (ARS), which shows mineralization processes through red depositions. 

Different plastic dishes were utilized for each staining. Cells were fixed with 4% paraformaldehyde and washed with phosphate-buffered saline (PBS). The stain was added, and the cells were incubated for 30 min. After lavage, the cells were observed with the phase-contrast microscope.

#### 2.2.4. Real-Time PCR

Total ribonucleic acid (RNA) was isolated using an SV Total RNA Isolation System kit. RNA from the amniotic stem cells of the control medium were used as the control group. A quantity of 1 µg of RNA was reverse-transcribed using a RETROscript kit.

Amplification was performed with specific primers for genes expressed during chondrogenic differentiation, such as type I collagen (COL I), type II collagen (COL II), chondroadherin (CNAD), cartilage oligomeric matrix protein (COMP), fibromodulin (FMOD), aggrecan (AGG), and osteogenic differentiation, such as bone sialoprotein (BSP), osteopontin (OPN), and for genes expressed in the mesenchymal stem cells considered to be markers of pluripotency (OCT-4 and SOX-2). The GAPDH gene was used as a reference for the standardization of the data (Table 1).

Amplifications were carried out using 35 cycles at 95 °C, 1 min; variable annealing temperature, 1 min; 72 °C, 1 min. RT-PCR products were separated in a 2% agarose gel and visualized via ethidium bromide staining. Images were captured using a Gel Doc 2000 (BioRad, Hercules, CA, USA).

## 3. Results

### 3.1. Molecular Characterization of the AFSCs

The AFSC lines used for our experiments, as previously described, were characterized by flow cytometry. As in previous studies, the AFSCs were positive for the mesenchymal stem-cell markers CD29, CD73, and CD44, while they were negative for the hematopoietic markers CD34 and CD45 and for the endothelial marker PECAM-1/CD31. Nevertheless, positivity for stemness markers (SSEA4, OCT4, Tra-1–60, and CD90) was found among the samples (data already published) [30].

### 3.2. Phase-Contrast Microscopy

After 4 days of coculture with PRP, it was possible to observe with a phase-contrast microscope round-shaped cell aggregates that had not been identified in the control medium. The cell aggregates increased in number and dimension until the 14th day (Figure 2a–c). 

There were no significant differences observed between the results of Protocol 1 and Protocol 2. 

The cells in the control medium maintained the fibroblastoid shape of the AFSCs for the duration of the experiment.

### 3.3. Staining 

On the 14th day of culturing, two different stains were investigated. The one using ABS, which binds sulfurized acid mucins and glycosaminoglycans, resulted as positive. In fact, the centers of the cell aggregates appeared blue using the phase-contrast microscope (Figure 3a–d).

The other one stain, using ARS, did not demonstrate any mineralization. Protocol 1 and Protocol 2 were similar in aspect and in staining. 

### 3.4. Real-Time PCR

The molecular study was performed using RT-PCR (Thermo Scientific Luminaris Color HiGreen qPCR Master Mix K0392). 

The decrease in the expression of OCT-4 in Protocols 1 and 2, as compared to the control, suggests the beginning of a differentiation process (Figure 4). 

The study of the specific markers for chondrogenic differentiation showed increased expressions of COL I, COL II, and COMP that were greater in Protocol 1, in which the PRP was activated through freeze–thaw cycles (Figure 5).

In regard to the specific markers for osteogenic differentiation (OPN and BSP), there were no changes in expression in Protocols 1 and 2, as compared to the control. 

## 4. Discussion

In recent decades, the use of PRP has gained popularity in the field of tissue engineering because of its anti-inflammatory and analgesic properties, as well as its capacity to induce neo-angiogenesis, tissue formation, and remodeling by influencing stem-cell migration, proliferation, and differentiation [14]. Because of these properties, the clinical use of PRP in the orthopedic field has increased more and more, with it finding an application with encouraging results in the most common orthopedic pathologies, such as muscle lesions, acute and chronic tendinitis, mild osteoarthritis, and osteochondral lesions [7,8,9,10,11,12].

If we focus on this latter pathology, the literature reports the use of PRP to induce the chondrogenic and osteogenic differentiation, both in vitro and in vivo, of MSCs harvested from several tissues [15,31,32,33].

While for in vivo usage the activation of PRP is not necessary, for in vitro applications, it is imperative [34], and several methods are available for PRP activation [6]. In our study, we selected two different strategies, and we decided to compare them with two protocols. In Protocol 1, PRP was activated using freeze–thaw cycles. In Protocol 2, PRP was activated through the addition of calcium gluconate.

For the choice of a platelet count in the PRP, we followed the recommendations of the Italian Society of Transfusion Medicine and Immunohematology (SIMTI) for PRP clinical use: 1 × 10^6^ μL ± 20%.

According to Lucarelli et al. [35], PRP has a dose-dependent effect on cell proliferation. A quantity of 10% PRP was proposed as a suggested concentration to promote the proliferation effect, while PRP concentrations of more than 20% have been demonstrated as having a negative impact [36].

On the other hand, AFSCs were chosen as the stem cells to be utilized, rather than more-commonly investigated alternatives. In fact, adult MSCs have limited potential and, even after reprogramming, they may maintain epigenetic modifications that limit their application. In regard to embryonic stem cells (ESCs), despite their high potential for differentiation, their use is associated with the risk of teratocarcinoma induction after in vivo transplantation and, furthermore, their harvesting gives rise to ethical problems due to the necessary suppression of the embryo [37]. Considering that, AFSCs can be seen as a very promising tool in the area of regenerative medicine.

AFSCs, in fact, represent an “intermediate” cellular phenotype between ESCs and adult MSCs; they express markers of both pluripotency and mesenchymal commitment, and they exhibit a broad differentiation potential for all three embryonic germ layers [37]. They can be a suitable cell source for tissue engineering, and their abilities in cartilage and bone defect repairs have been tested in established animal models [38,39]. 

In addition, AFSCs are nontumorigenic; they do not form teratomas in vivo when injected in immunodeficient mice [37], and the low immunogenicity of AFCSs supports their use in allogenic transplantation. In fact, AFSCs are slightly positive for the MHC class II antigens of HLA-DR, the expression of immuno-suppressive factors, such as CD59 (Protectin), which inhibits the complement system in damaging cells, and HLA-G, which plays a key role in immune tolerance in pregnancy, making them resistant to rejection [4].

However, despite these promising characteristics, in the international literature, only two works have investigated the association of AFSCs and PRP on tissue regeneration. Both authors used AFSCs premixed with PRP to restore bone defects in rats, showing encouraging results as compared to those of the control groups [40,41].

In addition to the previous mentioned varieties of MSC and PRP, in combination, they have been found to have several applications in orthopedic clinical practice for their synergistic effect, as broadly discussed. In regard to AFSCs, having seen their favorable properties, it would be desirable to see results from a clinical trial as soon as possible.

In the end, it is important to point out that we investigated the differentiation potentialities of PRP itself. In fact, to our knowledge, in most of the experiments performed to investigate the chondrogenic or osteogenic effect of PRP in vitro, PRP is usually added to chondrogenic or osteogenic media, or it is combined with biomaterials or recombinant growth factors. Therefore, the experimental designs do not allow conclusions to be drawn regarding the effect of the PRP itself [6].

Several limitations should be noted in regard to this study. First, there is a lack of biochemical or Western blot staining to confirm the data from the real-time PCR. The second limitation of this study is the absence of preclinical experiments on animal models, and one of our future priorities is to better investigate these results.

## 5. Conclusions

According to our data, looking at the phenotype observed with phase-contrast microscopy, the result of staining with ABS and ARS, and the results of RT-PCR, it can be assumed that adding PRP to a culture medium with AFSCs can influence them through chondrogenic differentiation.

Based on their ease of harvesting, their ability to differentiate into several cell lineages, the absence of tumorigenicity after transplantation, and the lack of ethical problems related to their use, AFSCs could be considered as a novel, promising resource in orthobiology. 

In conclusion, the association of PRP and AFSCs could be considered, in the foreseeable future, as a new tool in regenerative medicine for the treatment of chondral pathologies.

## Figures and Tables

**Figure 1 ijerph-19-15786-f001:**
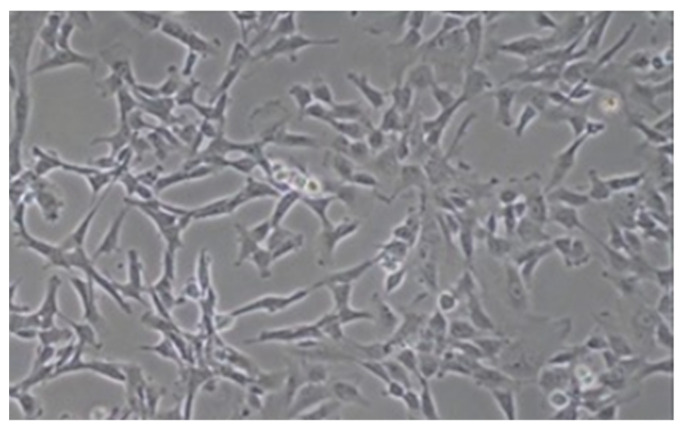
Third passage of the AFSC cell culture. They appear with a fibroblastoid-like shape (10× magnification with a phase-contrast microscope).

**Figure 2 ijerph-19-15786-f002:**
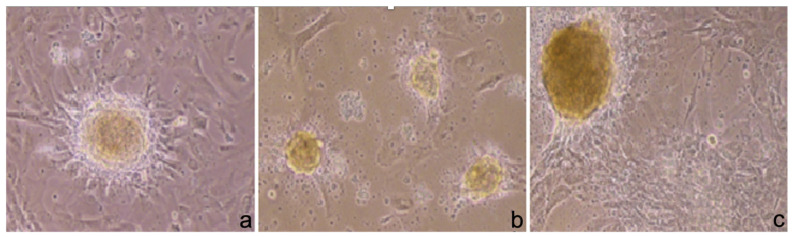
Cell aggregates in culture medium after 5 days (**a**), after 10 days (**b**), and after 14 days (**c**) (10× magnification with a phase-contrast microscope).

**Figure 3 ijerph-19-15786-f003:**
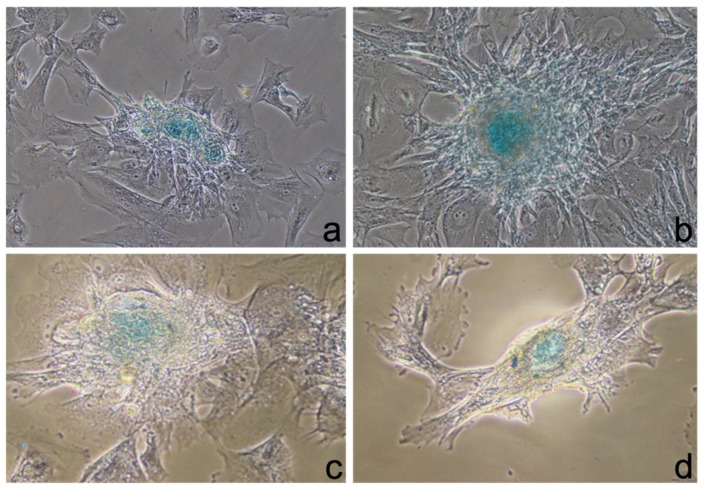
Alcian blue staining: (**a**,**b**) 10× magnification; (**c**,**d**) 20× magnification.

**Figure 4 ijerph-19-15786-f004:**
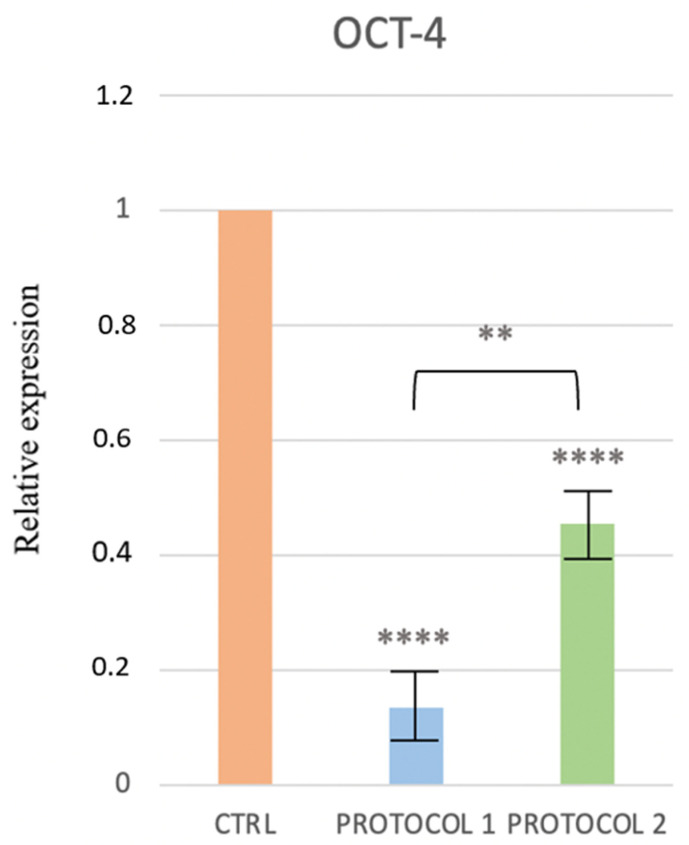
Change in the expression of OCT-4 gene in the control, Protocol 1, and Protocol 2. ** *p* ≤ 0.01 and **** *p* ≤ 0.0001.

**Figure 5 ijerph-19-15786-f005:**
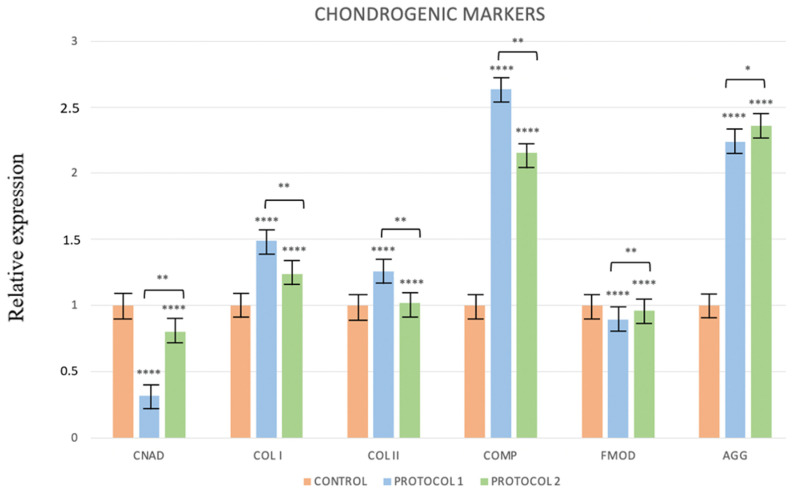
Change in the expression of chondrogenic markers in Protocols 1 and 2. * *p* ≤ 0.1, ** *p* ≤ 0.01, **** *p* ≤ 0.0001.

**Table 1 ijerph-19-15786-t001:** Primer sequences for pluripotency markers (1a), chondrogenic markers (1b), osteogenic markers (1c), and the control sequence (1d).

**1a**	
**PLURIPOTENCY MARKERS**	
OCT-4	Forward (5′ to 3′): CTT GCT GCA GAA GTG GGT GGA GGA
	Reverse (5′ to 3′): CTG CAG TGT GGG TTT CGG GCA
SOX-2	Forward (5′ to 3′): TTG CTG CCT TAA GAC TAG GA
	Reverse (5′ to 3′): CTG GGG CTC AAA CTT CTC TC
**1b**	
**CHONDROGENIC MARKERS**	
TYPE I COLLAGEN	Forward (5′ to 3′): CCA ATC ACC TGC GTA CAG AAC
	Reverse (5′ to 3′): GGC ACG GAA ATT CCT CCG GTT GAT
TYPE II COLLAGEN	Forward (5′ to 3′): CCA GGT CAA GAT GGT C
	Reverse (5′ to 3′): CTT CAG CAC CTG TCT CAC CA
CHONDROADHERIN	Forward (5′ to 3′): ACC TGG ACC ACA AGG TC
	Reverse (5′ to 3′): GAA CTT CTC CAG GTT GT
COMP	Forward (5′ to 3′): CAG GAC TTT GAT GCA GA
	Reverse (5′ to 3′): AAG CTG GAG CTG TCC TGG TA
FIBROMODULIN	Forward (5′ to 3′): ACC AGT GAT AAG GTG GGC AG
	Reverse (5′ to 3′): AAG TAG TTA TCG GGG ACG GT
AGGRECAN	Forward (5′ to 3′): GGC TTG AGC AGT TCA CCT TC
	Reverse (5′ to 3′): CTC TTC TAC GGG GAC AGC AG
**1c**	
**OSTEOGENIC MARKERS**	
OSTEOPONTIN	Forward (5′ to 3′): AGG AGG CAG AGC ACA
	Reverse (5′ to 3′): CTG GTA TGG CAC AGG TGA TG
BONE SIALOPROTEIN	Forward (5′ to 3′): CTA TGG AGA GGA CGC CAC GCC T
	Reverse (5′ to 3′): CAT AGC CAT CGT AGC CTT GTC CT
**1d**	
**CONTROL**	
GAPDH	Forward (5′ to 3′): CGC TCT CTG CTC CTG TT
	Reverse (5′ to 3′): CCA TGG TGT CTG AGC GAT GT

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
