# Peer review of "The Role of Platelet-Rich Plasma on the Chondrogenic and Osteogenic Differentiation of Human Amniotic-Fluid-Derived Stem Cells"

_ijerph, 2022, doi:10.3390/ijerph192315786_

Round 1

Reviewer 1 Report

1. The effect of bone regeneration in damaged tissues using platelet derivates and the enhancement of bone differentiation of MSCs have already been reported.

2. There is no data on the characterization of stem cells.

3. There are no error bars in all RT-PCR graphs, so the significant difference cannot be known.

4. RT-PCR alone is not sufficient to know the differentiation-promoting effect. Western blot, marker staining, and in vivo validation should be included.

5. Overall, it is a basic research and the results are very lacking.

Reviewer 2 Report

Dear Authors,

Thank you very much for opportunity to read this interesting article. “The role of platelet rich plasma on the chondrogenic and osteogenic differentiation of human amniotic fluid derived stem cells” by  Gianetti et al.

Below are  my suggestions : 

1)    The title should not be capitalized, but it is up to the editor to decide if such a version of the title can be.

2)    The abstract is unreadable, poorly written, and should be expanded. Suggests structuring the abstract.

3)    Introduction must be improved. You should mention more widely other clinical use of PRP, MSC, and amniotic fluids and amniotic membrane as well. Amniotic membrane is widely used as a biological tissue dressing and I recommend to extend a literature about this fact e.g. clinical applications in burns :  Consider this references : 

https://www.semanticscholar.org/paper/Biological-dressings-as-a-substitutes-of-the-skin-Gierek-Kawecki/35b802cba72781c5b1d58ae3244848ebc8eec1b6

4)    Figura 5 should be Figure – spell check is required

Discussion – you should improve this section. In my opinion more clinical applications rather than technical lab information.  You should be more narrative. What about shoulder injuries and PRP applications ? It is common injury, maybe you should consider this reference 

https://doi.org/10.3390/ijerph191911857

And I found interesting article about PRP in tennis elbow  :  https://doi.org/10.3390/jcm11123504

All in all, paper is very interesting, in my opinion technical bio-engineering with surgical specialties have great future. I suggest to make all improvements to reach the text more readeable for non-surgical readers as well.  Thank you so much for reading this very important manuscript.

Round 2

Reviewer 2 Report

Dear Authors,

Authors well adressed all suggestions. 

I think that this paper  :  https://doi.org/10.3390/jcm11216362

will definitely increase the value of reference section and you should mention this reference in discussion section as well.

Thank you very much, I will have no further comments. I think that authors corrected the manuscript according to the comments.

Author Response

Thank you for your further suggestion. We added the reference as the number 12.